# Multivariate Analysis of Protein–Nanoparticle Binding Data Reveals a Selective Effect of Nanoparticle Material on the Formation of Soft Corona

**DOI:** 10.3390/nano13212901

**Published:** 2023-11-04

**Authors:** Susannah Emily Cornwell, Sarah Ogechukwu Okocha, Enrico Ferrari

**Affiliations:** Department of Life Sciences, University of Lincoln, Lincolnshire, Lincoln LN6 7TS, UK

**Keywords:** protein corona, soft corona, dynamic light scattering, serum albumin, transferrin, prothrombin

## Abstract

When nanoparticles are introduced into the bloodstream, plasma proteins accumulate at their surface, forming a protein corona. This corona affects the properties of intravenously administered nanomedicines. The firmly bound layer of plasma proteins in direct contact with the nanomaterial is called the “hard corona”. There is also a “soft corona” of loosely associated proteins. While the hard corona has been extensively studied, the soft corona is less understood due to its inaccessibility to analytical techniques. Our study used dynamic light scattering to determine the dissociation constant and thickness of the protein corona formed in solutions of silica or gold nanoparticles mixed with serum albumin, transferrin or prothrombin. Multivariate analysis showed that the nanoparticle material had a greater impact on binding properties than the protein type. Serum albumin had a distinct binding pattern compared to the other proteins tested. This pilot study provides a blueprint for future investigations into the complexity of the soft protein corona, which is key to developing nanomedicines.

## 1. Introduction

When nanoparticles mix with blood, a plasma protein corona forms on their surface. This corona has been studied for over a decade to understand its effects on the efficacy of nanomedicines [1]. Protein coronas play a key role in the biodistribution [2,3], pathophysiology [4], targeting efficiency [5], cellular uptake [3,6,7,8] and toxicity of nanoparticles [7,8,9]. Predicting the exact protein composition of nanoparticle coronas in the blood is challenging due to the complex protein profile of plasma [10] and the variety of materials and chemical functionalizations used in nanomaterial synthesis [11]. However, mass spectrometry and standard proteomics approaches have been used to determine the protein corona composition for several nanomaterials [12,13].

To study the protein corona composition, nanoparticles must be separated from excess plasma using centrifugation or size-exclusion chromatography. During this process, loosely bound proteins, known as the “soft corona”, are lost. By contrast, the “hard corona” is formed by stably bound proteins that are analytically accessible [14,15]. There is an increasing focus on characterizing the soft corona, and new techniques such as biolayer interferometry [16], cryo-electron microscopy [17] and in situ click chemistry-mediated capture [18] have proved promising. However, characterizing the soft corona remains challenging [18,19], whereas the absolute quantification and identification of proteins in the hard corona can be routinely obtained and is highly informative for designing nanomedicines [20], although, due to the complexity of the experimental approach, inter-lab reproducibility is still an issue [21].

Nanoparticle protein coronas have a dynamic nature [22], and top-down characterization methods like mass spectrometry are seldom used to study their evolution over time [4]. An alternative approach is to use bottom-up computational methods to study protein–nanoparticle interactions over time. These methods can predict the composition of protein coronas when nanomedicines are exposed to different body fluids on their journey towards their intracellular targets [23,24]. Bottom-up methods use experimentally determined protein–nanoparticle binding affinities from single-protein solutions to calibrate computational models of combined plasma protein types adsorbed on nanoparticles. These models range from simple kinetic models based on association and dissociation rates of specific protein–nanoparticle pairs [25] to more complex coarse-grained molecular dynamics simulations [26]. Both model types assume that exposure to single-protein solutions leads to the formation of a densely packed monolayer of protein at the liquid–solid interface, with bound protein molecules occupying the entire surface area available on the nanoparticle. Computer-simulated protein mixtures matching native blood concentrations compete for the surface area available on nanoparticles using experimentally determined binding affinities. To validate these methods, end-point estimates of nanoparticle-adsorbed concentrations obtained from simulations are compared to experimental end-point concentrations measured in the hard corona of nanoparticles exposed to the actual mixtures for the same duration as the simulations.

Bottom-up computational methods for predicting protein corona composition have shown good agreement with experimental electrophoretic patterns obtained from simplified plasma protein mixtures made of three proteins [25,26]. However, to use these models in more complex scenarios, a better understanding of the “soft corona” is needed. The assumption that single-protein solution adsorption yields one monolayer of protein molecules may only be valid for the “hard corona”. Additionally, it may be difficult to design increasingly complex simulations and experiments without first understanding whether the nature of the nanomaterials or the diversity of the proteins involved has a greater impact on their interaction.

In this exploratory experimental study, we assessed and quantified the effect of nanoparticle materials and protein types on binding affinity and the thickness of the layer formed on nanoparticles incubated with a single-protein solution. We used dynamic light scattering (DLS) to measure particle size increase in response to protein adsorption, while simultaneously determining the thickness of the adsorbed protein layer and the dissociation constant [27,28]. Importantly, measurements were taken in the solution without the need for washing steps, allowing us to account for the soft corona.

We used multivariate analysis to compare the effects of two common nanomaterials, silica and gold nanoparticles, and three representative plasma proteins, serum albumin (SA), transferrin (TF) and prothrombin (PT), on the dissociation constant (*K_D_*) and maximum thickness (*τ_max_*) of the protein layer at saturation. We found a statistically significant interaction effect between material type and protein type on *K_D_* and *τ_max_*, but the overall effect was largely due to material type. We also found protein-specific binding properties: TF and PT had significantly higher affinity to gold than silica and formed a thick, soft corona on both materials, while SA did not show material-specific affinity and formed a thick, soft corona on gold, but only a thin monolayer on silica nanoparticles. These results highlight the importance of investigating both the soft and hard coronas to build a comprehensive understanding of the complex dynamics at the nanoparticle solid–liquid interface.

## 2. Materials and Methods

### 2.1. Materials

Serum albumin (SA), transferrin (TF) and prothrombin (PT) from human serum were purchased as lyophilized powders from Sigma-Aldrich (product code A8763, T3309 and 539515, respectively) and dissolved in 10 mM HEPES at pH 7.3 and 10 mM NaCl at approximately 10 mg/mL concentration. Their exact concentration was determined using the Bradford assay. A quantity of 100 nm silica nanoparticles was purchased from Polysciences as a 5% solid colloidal suspension in water, while 100 nm gold nanoparticles were purchased from NanoComposix as a 0.05 mg/mL colloidal suspension in 2 mM sodium citrate. The molar concentrations of the nanoparticles were calculated assuming a monodispersed spherical colloid. Details of the physical and chemical properties of each nanoparticle preparation were provided by the supplier and are reported in Appendix A.

### 2.2. Dynamic Light Scattering

All dynamic light scattering (DLS) measurements were performed using a Malvern Zetasizer Nano ZS. In these measurements, 150 µL solutions of protein–nanoparticle mixtures were equilibrated at 20 °C in disposable plastic micro-cuvettes and measured using a 173° back-scattering setup. The mean hydrodynamic diameter of the nanoparticles (*d*) and its standard error were determined from the Z-average of the scattering intensity size distribution over 4 repeats. The shift of nanoparticle diameter (Δ*d*) due to protein adsorption and the thickness of the protein corona (*τ*) were determined using Equation (1):(1)$$\tau =\frac{\Delta d}{2}=\frac{d-d_{0}}{2}$$

In Equation (1), *d_0_* represents the hydrodynamic diameter of control nanoparticles without protein. *τ* is measured over a range of protein concentrations *C* to determine the maximum thickness of the nanoparticle corona (*τ_max_*) at saturating protein concentration and the dissociation constant (*K_D_*) using the Hill–Langmuir model (Equation (2)):(2)$$\tau =\tau_{max}\frac{C}{K_{D}+C}$$

The 2-parameter Hill–Langmuir model of Equation (2) assumes no cooperative process in the adsorption (Hill coefficient equal to 1), as shown before [28,29]. This was verified by fitting the datasets with a 3-parameter model, which yielded Hill coefficients that were not significantly different from 1.

The concentration of 100 nm silica and gold nanoparticles used in the protein adsorption experiments was minimized to prevent any substantial depletion of the protein concentration in the solution upon binding. In these conditions, the total protein concentration used in the adsorption reaction (*C_0_*) is nearly equivalent to the unbound concentration *C* in Equation (2) and can be used in place of *C* when applying a Langmuir–Hill model to binding curves where proteins are titrated into a nanoparticle solution.

The expected thickness (*τ_ex_*) of a dense monolayer of adsorbed proteins was calculated from the diameter of a sphere approximating the size of a protein molecule according to Equations (3) and (4):(3)$$\tau_{ex}=2r$$

In Equation (3), *r* represents the radius of the sphere that approximates the shape of a globular protein in nm units, calculated using Equation (4) as previously reported [29,30]:(4)$$r=0.066\sqrt[{3}]{M_{r}}$$

In Equation (4), *M_r_* represents the relative mass of the protein in g mol^−1^. An *M_r_* value of 67,000, 80,000 and 72,000 g mol^−1^ was used for SA, TF and PT, respectively, as indicated by the supplier’s data sheets.

### 2.3. Statistical Analysis

All data were analyzed using the statistical computing software R version 4.2.2. The parameters *τ_max_* and *K_D_* of Equation (2) were fitted to the experimental data using non-linear least squares regression (R function “nls”). Multivariate analysis of variance (MANOVA) was performed using the R function “manova”, and the values of η^2^, measuring how much variance in *τ_max_* and *K_D_* is accounted for by the explanatory variables (nanoparticle material and protein), were calculated using the R function “eta_squared” from the “effectsize” library. Differences in *τ_max_* and *K_D_* estimates between individual groups (nanoparticle material and protein) were evaluated by computing the 95% confidence intervals using the R function “confint”. For non-overlapping confidence intervals, a significant difference (*p* < 0.05) was reported.

## 3. Results

The size and polydispersity index of a range of silica and gold nanoparticle concentrations mixed with 0.5 mg mL^−1^ serum albumin (SA) were measured via DLS to optimize the nanoparticle concentrations used in all the following binding experiments. The optimization aimed at verifying if the nanoparticle concentration affected the measurements of the hydrodynamic diameter d of nanoparticles in the presence of a high concentration of proteins, i.e., by yielding a polydisperse solution where scattering was contributed largely by the highly concentrated proteins rather than by the sparse nanoparticles. This may result in an apparent decrease in the overall average particle diameter in the presence of proteins, as previously observed for small nanoparticles having a size comparable to a protein [31].

The data in Appendix A show that the nanoparticle diameter measured in the absence of SA and the increment in size due to the adsorption of SA are consistent at any concentration tested and present low polydispersity for both silica and gold, confirming that DLS is relatively insensitive to nanoparticle concentration in the conditions used, most likely due to large-sized nanoparticles (100 nm) that scatter far more intensely than smaller proteins. A concentration of 0.4 nM and 2 pM was used for silica and gold nanoparticles, respectively, in all subsequent measurements. Both concentrations were chosen to be as low as possible for limiting protein depletion in the binding experiments and yet high enough to be reproducible given the volume of the test cuvettes and the concentration of the stocks.

The dissociation constant (*K_D_*) and maximum thickness of the protein layer (*τ_max_*) for nanoparticle–protein pairs were determined by fitting the adsorption isotherms of Figure 1 to a Langmuir–Hill model. Binding curves were obtained by mixing 100 nm silica nanoparticles with varying concentrations of SA (Figure 1a), transferrin (TF, Figure 1b) and prothrombin (PT, Figure 1c). In addition, SA, TF and PT were titrated into solutions containing 100 nm gold nanoparticles (Figure 1d–f, respectively). Protein concentrations ranged from 0 to 0.5 mg mL^−1^ for silica nanoparticles and from 0 to 0.2 mg mL^−1^ for gold nanoparticles to account for the lower dissociation constants observed in the latter.

A summary of the parameters obtained from fitting the data in Figure 1 is presented in Table 1. The results suggest that, in general, *K_D_* is lower for gold nanoparticles, except SA, for which a similar *K_D_* was obtained for both materials. *τ_max_* does not differ significantly between the materials, except for SA, where a notably smaller thickness was observed for silica nanoparticles.

Figure 2 shows a scatter plot of *K_D_* and *τ_max_*, highlighting that the data presented in Table 1 appear to cluster based on the nanoparticle material. In this context, a cluster is defined as a set of data points that occupies a distinct area of the scatter plot without overlapping with another set. To support this observation, multivariate analysis of variance (MANOVA) was applied to the data in Table 1 to assess the interaction between the two independent variables, protein type and nanoparticle material, in determining the outcomes of the two dependent variables, *K_D_* and *τ_max_*. The Pillai statistic from the multivariate test was lower than 10^−16^ for both protein type and nanoparticle material, indicating that both independent variables yielded statistically different outcomes. Notably, the Pillai statistic for the interaction of the two independent variables was also lower than 10^−16^, indicating a statistically significant interaction between them. To determine the effect size of each independent variable on the combined variance of *K_D_* and *τ_max_*, we calculated η^2^, where a value from 0 to 1 corresponds to an effect size from small to large. The η^2^ of nanoparticle material was 0.93, while protein type had an η^2^ of 0.59. The interaction of nanoparticle material and protein type yielded an η^2^ of 0.69. These effect size values suggest that nanoparticle material has the largest effect, and the moderately large η^2^ for the interaction also highlights material-specific binding properties for the protein types tested.

To examine the effects of protein type and nanoparticle material on *K_D_* and *τ_max_* in detail, these variables are represented independently in the bar charts of Figure 3 and Figure 4, respectively. Significant differences are highlighted based on the lack of overlap of the confidence intervals calculated in Table 1.

Figure 3 illustrates that SA presents significantly lower *K_D_* and *τ_max_* for silica nanoparticles, while these values are significantly higher for gold nanoparticles, compared to TF and PT. The binding properties of TF and PT are more similar to each other. Figure 4 presents the same data as Figure 3, but the data are grouped by protein type to emphasize the effects of nanoparticle materials on individual protein types.

Figure 4a shows that the *K_D_* of TF and PT is much lower, indicating higher affinity, for gold nanoparticles compared to silica nanoparticles. However, in the case of SA, there is no significant difference in *K_D_* between the two nanoparticle materials. On the other hand, while the *τ_max_* of TF and PT is similar for both nanoparticle materials, SA forms a significantly thinner protein corona on silica nanoparticles compared to gold (Figure 4b).

In Figure 4b, the *τ_max_* values were compared to the expected thickness *τ_ex_* for a protein monolayer. The exact arrangement and space occupied by the adsorbed proteins is unknown, but the measured thickness was compared to a monolayer formed by proteins approximated to spheres packed at the maximum density on the nanoparticle surface. In this configuration, *τ_ex_* is equivalent to the diameter of the sphere approximating SA, TF and PT, which was calculated based on existing models, resulting in values of 5.4, 5.7 and 5.5 nm for SA, TF and PT, respectively [29,30]. The data in Figure 4b show that *τ_max_* is thicker than *τ_ex_* in all cases, except for SA on silica nanoparticles, suggesting that the protein soft corona of the single-protein mixtures presents multiple layers of protein in all cases, except for SA on silica nanoparticles, where the estimated *τ_max_* value is compatible with a monolayer.

## 4. Discussion

In protein–nanoparticle binding experiments that focus on the soft corona and potentially loosely bound protein molecules, the concentration of nanoparticles needs to be as low as possible to minimize protein depletion in the solution so that the total concentration of protein (*C_0_*) can be assumed to be similar to the unbound concentration (*C*) and a Hill–Langmuir model can be applied. The bound and unbound protein concentrations cannot be measured without the separation of nanoparticles from the solution, which would affect the soft corona; therefore, it is only possible to rely on minimizing the nanoparticle concentration so that *C_0_ ≈ C*. Appendix A illustrates this concept by plotting the binding curve of Figure 1a, taking into account protein depletion at different nanoparticle concentrations. This condition is equivalent to minimizing a receptor concentration when studying the binding of a ligand in biochemistry. If the nanoparticle concentration is too low, however, DLS measurements of protein adsorption on nanoparticles may be inaccurate due to excessive contribution from protein molecules in the solution to overall scattering [31]. We mitigated the latter by using relatively large nanoparticles (100 nm), whose scattering is substantially higher than the scattering of ~5 nm proteins (the scattering intensity is proportional to the sixth power of the size). The optimal nanoparticle size and nanoparticle concentrations used in this study were key to producing truly representative binding curves and allowing subsequent analysis.

The analysis of binding data for the two nanoparticle materials and three protein types reveals that both variables impact the adsorption properties, with the nanomaterial type being the primary source of variance. Blood plasma contains more than 1000 different proteins [10], and over 100 have been reported to form a hard protein corona on silica [4] and gold nanoparticles [32]. Therefore, the scale of the experiments reported here is not sufficient to cover the vast diversity of the blood plasma proteome. Instead, the intent was to validate a method that can highlight certain aspects of protein corona complexity, such as determining the effect size of nanoparticle materials and protein types, and identifying the unique binding properties of individual proteins.

Our study focused on three proteins: serum albumin (SA), transferrin (TF) and prothrombin (PT), which can be easily sourced and have comparable biochemical properties, as shown in Table 2. Despite similarities in terms of relative mass, isoelectric point (pI), and hydrophobicity (Gravy index), we observed a notably different set of binding properties for serum albumin. Previous studies have shown that simple biochemical properties cannot predict the binding and relative abundance of the nanoparticle protein corona in the case of the hard protein corona [13], and our experiments support the hypothesis that this is also true for the soft corona. It is likely that amino acid composition and structural properties contribute to the variance in binding properties and interact with the variance due to the specific nanoparticle materials tested. Specific protein structure–nanomaterial interactions have been described in a limited number of cases and for a limited interaction timescale using molecular dynamics simulations for silica [29], gold [28,33] and other relevant surfaces [34].

Unlike proteomics studies, which use mass spectrometry to identify and quantify the relative abundance of proteins within the nanoparticle hard corona, our work focused on the soft corona, which is likely to be more physiologically relevant [37]. We used dynamic light scattering (DLS) to determine nanoparticle size and, by titrating proteins into the nanoparticles, to provide an accurate estimate of the dissociation constant (*K_D_*) and maximum thickness of the protein layer (*τ_max_*) in a label-free and in-solution manner. DLS has been previously applied to study the binding of individual proteins [38], blood serum [31], or both [39] to nanoparticles. Other label-free methods used to determine the binding properties of protein–nanoparticle pairs include microscale thermophoresis [26], isothermal titration calorimetry [39] and tryptophan fluorescence [40]. However, unlike these techniques, DLS can directly estimate the thickness in nm of protein soft coronas, allowing speculation on the packing of single-protein soft coronas on nanoparticles. Our data suggest that proteins generally over-pack within the soft corona. However, SA appears to adsorb onto silica nanoparticles, forming a thinner corona that is compatible with a monolayer, as previously observed for bovine serum albumin at neutral pH [40].

A limitation of DLS is that it would not be possible to distinguish the contribution to the soft corona thickness by different protein types when mixtures of proteins are used instead of individual proteins. Fluorescence quenching (FQ) or fluorescence correlation spectroscopy (FCS) have been applied to study the kinetics of protein–nanoparticle interaction [14,41,42,43,44]. Due to the range of fluorophores available, these techniques could be potentially used to simultaneously measure the binding properties of more than one protein type. However, whereas DLS is a label-free method, labeling proteins with fluorophores for FQ or FCS requires cross-linking via thiol, amino or carboxylic groups presented by multiple protein residues, which is potentially invasive and may affect the protein properties.

Although the binding of single-protein solutions to nanoparticles may not be directly comparable to their actual adsorption within a complex mixture of plasma proteins, information on the interaction of nanoparticle–protein pairs is invaluable for calibrating increasingly accurate models to describe the dynamic nature of protein coronas [45,46,47].

This study focused on the multivariate analysis of protein–nanoparticle interaction, taking into account two variables, protein and material types, due to the fact that the chemical properties of biomolecules, the hard material that comprises the nanoparticles, and any chemical modification on the nanoparticle surface are likely to be the main drivers of the variability observed in the interactions. In more complex scenarios, it is possible that the physical properties of the nanomaterials used, e.g., nanoparticle size and shape, can also affect the formation of the soft corona and may need to be included in a more comprehensive multivariate analysis. The effect of size and shape on the formation of a protein corona has been studied before, primarily focusing on the hard corona, and it has been recently reviewed [48]. Both size and shape determine the curvature. Smaller spherical nanoparticles present greater curvature and potentially less contact area with proteins, yielding thinner coronas, although this becomes evident only with very small nanoparticle sizes (<10 nm) [31]. Similarly, nanoparticle shapes that better complement the shape of a protein type will provide more contact areas and promote the formation of a protein corona. It is important to point out that different protein types will present different flexibility in their polypeptide chain/s and will have a different ability to adapt to the size and shapes of the nanomaterial they are interacting with. Generally, a very small nanoparticle or a nanoparticle whose shape increases the overall curvature of the surface will preferentially recruit flexible proteins compared to a large, lower curvature nanosphere [48]. This highlights a likely crosstalk between variables, such as protein type, nanoparticle size and nanoparticle shape, emphasizing once more the importance of a multivariate analysis approach.

## 5. Conclusions

In summary, our multivariate analysis of the binding properties of serum albumin, transferrin and prothrombin onto silica or gold nanoparticles reveals that the material of the nanoparticles is a larger source of variance in the binding parameters compared to the different protein types. However, there is also a significant interaction between nanoparticle type and protein type. A detailed investigation shows that the binding pattern of serum albumin is unique compared to the other proteins tested. All proteins were able to form an extensive soft corona on nanoparticles, with a thickness suggesting over-packing. The only exception was serum albumin adsorbed on silica nanoparticles, which appeared to form a dense monolayer instead.

A detailed understanding of the protein soft corona, including the impact of different nanomaterials on it, is a key step toward a better design of nanomedicines and the ability to predict their properties.

## Figures and Tables

**Figure 1 nanomaterials-13-02901-f001:**
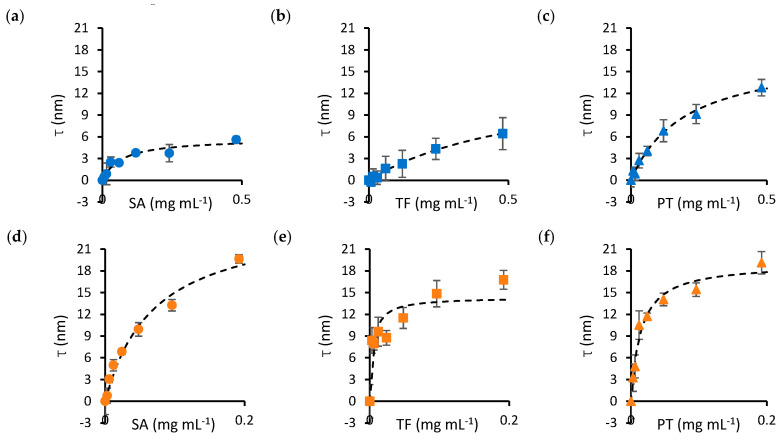
Binding curves (τ versus protein concentration) of protein–nanoparticle mixtures. The top panels show the adsorption isotherms obtained by mixing SA (**a**), TF (**b**) and PT (**c**) with silica nanoparticles. The bottom panels show SA (**d**), TF (**e**) and PT (**f**) mixtures with gold nanoparticles. The dashed curves on each panel represent the fits of the experimental data to the Langmuir–Hill equation, from which the parameters in Table 1 have been computed. Error bars represent the standard error of the mean over four repeats. Circle markers identify SA, square markers identify TF and triangle markers identify PT. Blue markers indicate data obtained for silica nanoparticles, while orange markers identify gold nanoparticles.

**Figure 2 nanomaterials-13-02901-f002:**
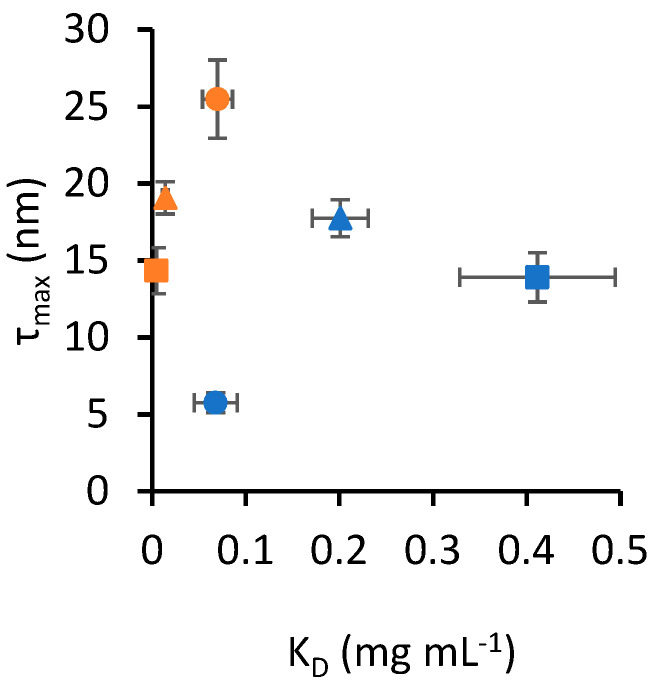
Scatter plot of *K_D_* and *τ_max_* values from Table 1. Error bars represent the standard error of estimates. Circle markers identify SA, square markers identify TF and triangle markers identify PT. Blue markers indicate data obtained for silica nanoparticles, while orange markers identify gold nanoparticles.

**Figure 3 nanomaterials-13-02901-f003:**
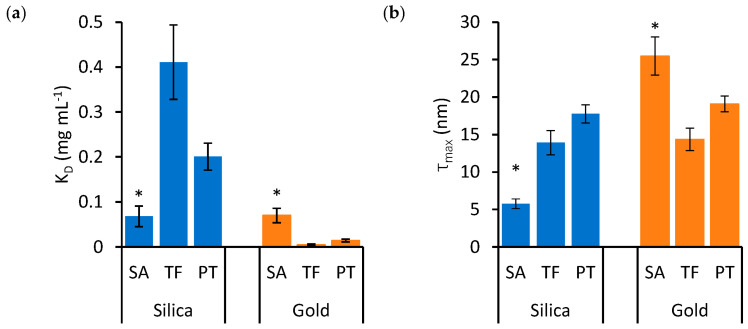
The effects of protein type on *K_D_* (**a**) and *τ_max_* (**b**) in mixtures of silica (blue bars) and gold (orange bars) nanoparticles. Error bars represent the standard error of estimates and asterisks identify groups that are statistically different from the others (* *p* < 0.05).

**Figure 4 nanomaterials-13-02901-f004:**
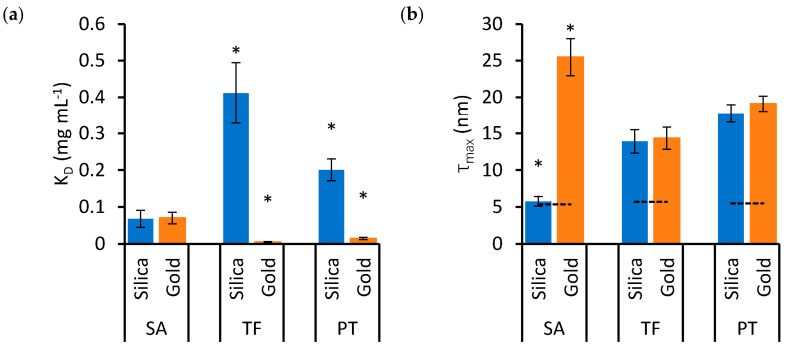
The effects of nanoparticle material on *K_D_* (**a**) and *τ_max_* (**b**) in SA, TF and PT mixtures. Blue bars indicate silica nanoparticles, while orange bars indicate gold nanoparticles. Error bars represent the standard error of estimates and asterisks identify groups that are statistically different from the others (* *p* < 0.05). The dashed lines in (**b**) represent the value of *τ_ex_*, which is the expected thickness for a monolayer of the relevant protein.

**Table 1 nanomaterials-13-02901-t001:** Estimates of *K_D_* and *τ_max_* obtained from the binding isotherms of Figure 1. se represents the standard errors of the estimates.

Protein Type	Nanoparticle Material	*K_D_* ± se (mg mL^−1^)	*τ_max_* ± se (nm)
SA	Silica	0.068 ± 0.023	5.8 ± 0.7
	Gold	0.070 ± 0.016	25.5 ± 2.5
TF	Silica	0.411 ± 0.083	13.9 ± 1.6
	Gold	0.005 ± 0.002	14.3 ± 1.5
PT	Silica	0.201 ± 0.030	17.8 ± 1.2
	Gold	0.014 ± 0.003	19.1 ±1.1

**Table 2 nanomaterials-13-02901-t002:** Biochemical properties of the proteins tested. The isoelectric point (pI) and the Gravy index were calculated using ProtParam [35] and the protein sequence available under the indicated UniProt ID [36]. The relative mass (M_r_) was obtained from the supplier of the proteins.

Protein Type	UniProt ID	M_r_ (kDa)	pI	Gravy Index
SA	P02768	67	6.67	−0.395
TF	P02787	80	6.70	−0.411
PT	P00734	72	5.23	−0.606

## Data Availability

The data presented in this study are available from the corresponding author upon request.

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
