# Peer review of "Multivariate Analysis of Protein–Nanoparticle Binding Data Reveals a Selective Effect of Nanoparticle Material on the Formation of Soft Corona"

_nanomaterials, 2023, doi:10.3390/nano13212901_

Round 1

Reviewer 1 Report (Previous Reviewer 1)

Comments and Suggestions for Authors

The authors have substantially improved the quality of their manuscript, but there are some issues that should be addressed before publication

-       The authors used the Hill-Langmuir model to describe the protein adsorption process on the nanoparticle surface. However, this model also contemplates the Hill coefficient, but the authors did not mention it. Is it 1 for all studied systems? Is it possible to exclude a cooperative process?

-       The authors, who aimed to study the soft corona, correctly reported the thickness of the protein corona by measuring the hydrodynamic diameter without separating the nanoparticles from the excess of proteins. However, I think that could be very useful also comparing these data with the thickness of the hard corona.

-       In the introduction, it might be useful to add some references highlighting the key role of nanoparticle surface properties in determining the extent and composition of the protein corona and, consequently, their biological activity. I can suggest the following references: Small 2018, 14, 1801219; Nanoscale, 2021, 13, 14119; J. Am. Chem. Soc. 2012, 134, 2139

Author Response

Reviewer 2 Report (New Reviewer)

Comments and Suggestions for Authors

I have carefully reviewed the paper titled "Multivariate analysis of protein-nanoparticle binding data reveals a selective effect of the nanoparticle material on the formation of soft corona." While the topic is certainly promising and relevant, I regret to inform you that I cannot recommend it for publication in its current form. Below are my specific comments and suggestions to help you improve your work.

1.Lack of Innovation and Experimental Validation:

Your paper lacks innovation, and it is evident that there is a dearth of experimental data. Furthermore, there is a conspicuous absence of confirmatory experiments to support your findings. It is critical to substantiate your theoretical approach with practical results to bolster the paper's credibility. You should consider expanding your experimental section and ensuring that the data is rigorously verified.

2.Doubts about the Use of DLS:

One of the most significant concerns in your study is the utilization of Dynamic Light Scattering (DLS) to investigate protein corona thickness. Numerous studies have shown that DLS measurements of protein corona thickness can be imprecise. This inaccuracy is due to the effects of hydrophilicity and hydrophobicity on nanoparticles after protein adsorption, which can alter the nanoparticles' interaction with DLS and, consequently, its outcomes. To address this issue, I strongly recommend reviewing the work of Warren Chan and similar researchers. It is imperative to select a technique that is well-suited to the characteristics of protein corona thickness. 

3.Outdated Background Research:

Your background research appears to lack the most recent advancements in the field. It is essential to incorporate the latest studies and methodologies for soft corona analysis, especially as this is a rapidly evolving area of research. Please consider referencing and citing recent articles such as those in Nature Communications, specifically "Nat Commun 2020, 11 (1),5389," " Nat Commun 2022, 13, 5389." and "Nat Commun 2021, 12 (1), 573." This will lend more current context to your paper and demonstrate your awareness of the field's progress. 

4. Importance of Understanding Interactions between Proteins:

Soft corona is not solely about the interaction between proteins and nanoparticles. Numerous experiments have already demonstrated that the interactions among proteins within the soft corona play a significant role. Therefore, the paper requires further research to elucidate the interactions between proteins, as this aspect remains an essential dimension of soft corona analysis

5.Need for More Experimental Validation:

While the use of computer simulations to predict protein corona behavior is commendable, your paper still requires more substantial experimental validation and a demonstration of its practical significance. Incorporating additional empirical evidence would enhance the overall quality of your research.

 In conclusion, I encourage you to address the issues mentioned above and revise your paper accordingly. The topic of soft corona in protein corona formation is indeed promising, and with more rigorous experimentation and a deeper understanding of measurement techniques, your research can significantly contribute to this field.

Round 2

Reviewer 1 Report (Previous Reviewer 1)

Comments and Suggestions for Authors

The authors addressed the issues raised

Reviewer 2 Report (New Reviewer)

Comments and Suggestions for Authors

Authors made detailed revision as suggestion which dramatically improve the quality. I aggree tha it is acceptable.

This manuscript is a resubmission of an earlier submission. The following is a list of the peer review reports and author responses from that submission.

Round 1

Reviewer 1 Report

Comments and Suggestions for Authors

In this study, an experimental exploration is conducted to understand how different types of nanoparticles and proteins interact in a solution. The researchers employed dynamic light scattering (DLS) to measure changes in particle size when proteins adsorb to the nanoparticles. This allowed them to evaluate the binding affinity and thickness of the protein layer. Two common nanoparticle materials, silica, and gold nanoparticles, were tested, along with three representative plasma proteins (serum albumin, transferrin, and prothrombin).

It is my opinion that this work is based on limited data and that even the available data is insufficient. I believe that additional experiments should be performed using more complex systems (such as combinations of proteins) and with additional data, like the variation of potential.

For these reasons, I think this article is not mature enough to be published.

Below are some addition comments:

- There is a lack of information about the nanoparticles; what are they coated with? What is their surface charge?

- From observing the graphs presented in Figure 1, it does not appear that saturation concentration is truly reached in several cases.

- The data and associated errors should be reported with the appropriate significant figures, nothing more.

- The authors state that 'Figure 2 shows a scatter plot of KD and τmax, highlighting that the data of Table 1 appear to cluster based on the nanoparticle material,' but it doesn't seem that they are truly clustered, at least for the silica nanoparticles.

- Figure 2, Figure 3, and Figure 4 present the same data repeatedly; I believe it is excessive to dedicate so much space to the same data, and some figures should be included in the supplementary material.

- Figure 5 does not serve a significant purpose.

Comments on the Quality of English Language

The quality of the English in this article is good

Reviewer 2 Report

Comments and Suggestions for Authors

This manuscript applied dynamic light scattering to determine the thickness of the protein corona absorbed on silica and gold nanoparticles. Experimental results of different plasma protein species and concentrations were also compared and discussed. However, the data and experimental details presented are insufficient for the conclusions drawnControl of the ratio of nanoparticles and protein molecules within the incubation systems are very important and have to be consideredThese comparisons would be inappropriate withoutrigorous physical and chemical characterizations ofnanomaterials used in this study. Furthermore, thispaper is brief and relies on a small amount of dataIt is difficult to understand the expected thicknesssince the explanation is unclear.

Comments on the Quality of English Language

The English is good, and the paper is of straightforward reading.

Reviewer 3 Report

Comments and Suggestions for Authors

Authors used DLS to determine the dissociation constant and thickness of the protein corona in solutions of silica or gold spherical nanoparticles mixed with serum albumin, transferrin, or prothrombin. They conclude that the nanoparticle material had a greater impact on binding properties than the protein type. Serum albumin had a distinct binding pattern compared to the other proteins tested. This work is thorough and extensive. However, whether or not there are differences in the difficulty of forming protein corona varies among nanoparticles with different morphologies. Please predict and explain.